# An interpretable machine learning algorithm enables dynamic 48-hour mortality prediction during an ICU stay
Simone Britsch [1,2,9] ✉, Markward Britsch [1,2,3,4,5,6,9], Simon Lindner[1,2], Leonie Hahn[1,2], Verena Schneider-Lindner[7,8], Thomas Helbing[1,2], Manfred Thiel[7,8], Daniel Duerschmied [1,2] & Tobias Becher[1,2]

## Abstract

**Background** Accurate short-term mortality prediction is essential for optimizing ICU management and improving patient outcomes. Many existing models rely on static data and do not reflect the dynamic progression of critical illness. This study aimed to develop and validate an interpretable machine learning algorithm that enables dynamic 48-hour mortality prediction throughout the ICU stay.

**Methods** We conducted a retrospective cohort study using electronic health records of 9,786 ICU patients treated between 2018 and 2022 at a German university hospital. A machine learning model was developed to predict 48-hour mortality, updated every 24 hours during the ICU stay. We trained and evaluated a Light Gradient-Boosting Machine using nested cross-validation and assessed performance via area under the receiver operating characteristic curve. External validation was performed on the MIMIC-IV database. Feature importance was analyzed using SHAP values.

**Results** Here, we show that the Light Gradient-Boosting Machine algorithm (LGBM-48h) achieves AUROCs of 0.909 (95% CI: 0.901–0.917) in the training and 0.886 (95% CI: 0.878–0.895) in the testing dataset. External validation using the MIMIC-IV database yields an AUROC of 0.859 (95% CI: 0.849–0.870). The model enables effective risk stratification across the ICU stay and reflects individual changes in patient status over time. Time-varying SHAP values improve interpretability by highlighting associated features.

**Conclusions** LGBM-48h provides a dynamic and interpretable framework for short-term ICU mortality prediction. The model may support clinical decision-making and prioritization of care, but requires further validation in real-time and prospective settings.

## Plain Language Summary

In this study, we developed a computer algorithm to help doctors predict whether a patient in the intensive care unit (ICU) may die within the next 48 hours. This can support timely treatment decisions and improve patient care. We used medical records from nearly 10,000 ICU patients to train the algorithm and tested it on data from another hospital. The algorithm updates its prediction every day using new patient information. We found that the tool was able to reliably identify high-risk patients and adapt to changes in their condition. It also shows which clinical values most influence the risk. This kind of technology could help ICU teams better plan treatments and use resources, especially in very busy hospital environments.

Intensive care units (ICUs) are specialized departments that offer the highest level of care for critically ill patients. Given the substantial investments required to establish and maintain ICU beds, their capacities are inherently limited. ICU overcrowding and resource strain have been identified as significant contributors to poor patient outcomes[1–3]. The recent COVID-19 pandemic has again highlighted the critical impact of ICU overcrowding and the resultant capacity constraints on patient care, emphasizing the necessity for efficient resource allocation in ICUs[4–6].

The accurate prediction of ICU outcomes may aid in clinical decision-making and the effective allocation of resources. A low predicted mortality can serve as the basis for a decision to discharge, while a high predicted

mortality may require further monitoring and care. Various scoring systems, such as the Sequential Organ Failure Assessment (SOFA) score, the Simplified Acute Physiology Score II (SAPS-II), the Acute Physiology and Chronic Health Evaluation (APACHE) score, the Logistic Organ Dysfunction Score (LODS), and the Oxford Acute Severity of Illness Score (OASIS), have been developed and are used clinically to predict mortality[7–11]. However, most of these traditional scores are static, considering only values at admission without capturing the dynamic changes throughout an ICU patient's stay. Additionally, ICUs generate vast, complex datasets that conventional scoring systems do not fully encapsulate, with data often changing abruptly and requiring interpretative support.

The advent of electronic health records (EHRs), comprehensive monitoring, and real-time data availability, combined with advanced data analytics, has facilitated the development of novel machine-learning-based algorithms[12,13]. Machine learning models for ICU prediction range from tree-based methods such as LightGBM to deep learning architectures, such as long short-term memory (LSTM) networks. While LSTM networks are capable of modeling temporal dependencies, they are often less interpretable and require higher computational resources. In contrast, gradient boosting tree algorithms offer strong predictive performance with lower complexity and greater transparency, which are important prerequisites for real-world clinical implementation[14]. Several dynamic, longitudinal models have been created to predict long-term and in-hospital mortality using both short-term (1 hour and below) and long-term (24 h) data sampling[15,16]. While these algorithms have shown excellent performance, some only apply to parts of the ICU stay[13], do not provide transparency regarding which features contribute to the model's outcomes[16–18], and thus, their real-time clinical utility is limited. Recently, an ensemble-based machine learning (ML) algorithm that employs a minimal set of common clinical variables from EHRs and can be applied hourly has bridged this gap by predicting 24 h mortality longitudinally throughout an ICU stay[19]. Although this algorithm has demonstrated excellent predictive performance in external, international datasets, a framework for effectively communicating outcome predictions to clinicians to support their translation into practice is still lacking.

In this study, we present a machine learning-based prediction model that estimates the risk of ICU mortality within the next 48 h, updated daily throughout the ICU stay. The algorithm shows consistently strong predictive performance across the entire ICU trajectory and in various diagnostic subgroups. It enables dynamic risk stratification and highlights individual changes in patient status. Our approach provides a framework for implementing interpretable, time-updated mortality prediction in intensive care, with potential for clinical translation.

## Methods

### Study design and ethics approval

This investigation is a retrospective, observational cohort study encompassing all patients aged 18 and above who received treatment in an ICU at the University Medical Center Mannheim, Germany, between January 2018 and May 2022. The study adhered to the principles outlined in the Declaration of Helsinki and received approval from the Medical Ethics Commission II of the Faculty of Medicine Mannheim, University of Heidelberg, Germany (Institutional Review Board approval number 2023-8990). Due to the study's retrospective design, the requirement for informed consent was waived.

### Data collection and ICD codes

All patient data available during ICU stays were automatically recorded in a patient data management system (PDMS). This dataset included vital parameters, medications, laboratory results, treatments, and outcome metrics. The data were provided by the Data Integration Center of the Medical Center Mannheim, Germany as a relational database, and specific data points were extracted using structured query language (Fig. 1).

The German modification of the International Classification of Diseases codes, 10th revision (ICD-10)[20] representing working diagnoses at hospital admission were assigned by the treating physician. ICD-10 codes indicating diagnoses at the end of the hospital stay were determined after a comprehensive review of post-discharge information and were obtained from the Department of Medical Controlling at the Medical Center Mannheim, Germany. "Admission diagnosis" refers to the diagnosis assigned on hospital admission, whereas "final diagnosis" (also "main diagnosis") refers to the primary diagnosis assigned after review of all data as described above. A summary of available data and ICD-10 codes is provided in Table 1 and Supplementary Table 1. An overview of all inclusion and exclusion criteria applied for patient selection is provided in Supplementary Table 10.

### Variable inclusion and definitions

From the available data, a panel of clinical experts identified variables for study inclusion along with their corresponding valid ranges. Variable selection was guided by the availability of data throughout the ICU stay, the association of variables with outcomes in the ICU setting, and the representation of all relevant organ systems in critically ill patients. The selected variables were categorized as either static (i.e., those that do not change during the ICU stay, such as age, source of admission, body mass index (BMI), and sex) or dynamic (i.e., those that change during the ICU stay, including vital parameters and lab values). All available variables, defined valid ranges and the percentage of measurements outside of the valid range that were excluded (i.e., outliers) are presented in Supplementary Table 2. To define time intervals for analysis, we divided each ICU stay (referred to as patient stay) into blocks of 24 h (referred to as stay day) starting on the time of ICU admission (Fig. 1).

Medications were represented as continuous variables, defined by the total dose of a substance administered within a single stay day. Ventilator therapy was quantified by the total duration of ventilation since the last initiation. Fluid therapy was recorded both as the fluid balance for each stay day and for the entire patient stay. Renal replacement therapy was encoded as a Boolean variable, indicating either the presence or absence of renal replacement therapy within one stay day.

For each of the measured continuous variables, aggregate variables based on the number of available measurements for each stay day were calculated. For variables with a median number of measurements per stay day smaller than ten, the median, minimum, arithmetic mean, and maximum values were calculated. For variables with ten or more measurements per stay day, additionally, the first quartile, the third quartile, and the standard deviation were calculated. In case the number of valid measurements for a variable was zero (or below two for standard deviation), the aggregated variable was considered missing. All available static and dynamic categorical and continuous aggregate variables were selected as features for subsequent ML algorithm development, resulting in a total of 131 features (Supplementary Table 3).

### Missing data and imputations

A patient's stay was excluded if over 30% of the features were missing during the initial 24-hour period. Likewise, a stay day was excluded if more than 30% of the features were missing. Incomplete stay days, defined as those with a duration of less than 24 h, were also excluded.

To impute missing features, patient stays were categorized into nine main ICD-10 disease groups based on admission diagnosis (Supplementary Table 1). Missing features within the first 24 h of an ICU stay were imputed using the aggregated feature variables from the corresponding ICD-10 disease group. For all following stay days, we utilized the most recent feature from earlier stay days within the same ICU stay (Fig. 1).

### Feature processing and difference features

Unordered categorical features were converted into dummy features. Ordinal variables were encoded as ordered factors for the LightGBM (LGBM, Light Gradient-Boosting Machine) and Random Forest algorithms, and as integers for other methods. A shifted logarithm transformation, $(f(x) = \ln(x + 1))$, was applied to features that were highly skewed towards zero values (Supplementary Table 3).

For each feature, a missingness indicator was included (total of 34 features). Additionally, a separate dataset was created, containing indications of the difference from the previous day's feature (if available) for each feature that was not constant. This dataset excluded the first day of the patient's stay. The final outcome of the ML algorithms on the test dataset was generated using two models: one trained on a training dataset without difference features (consisting of 131 features) and the other trained on a training dataset that included difference features (consisting of 248 features). The first model was used to predict the initial 24 h period, while the second model was utilized to predict the remaining days of the stay. The model was designed to generate predictions once every 24 h, with a 48 h mortality

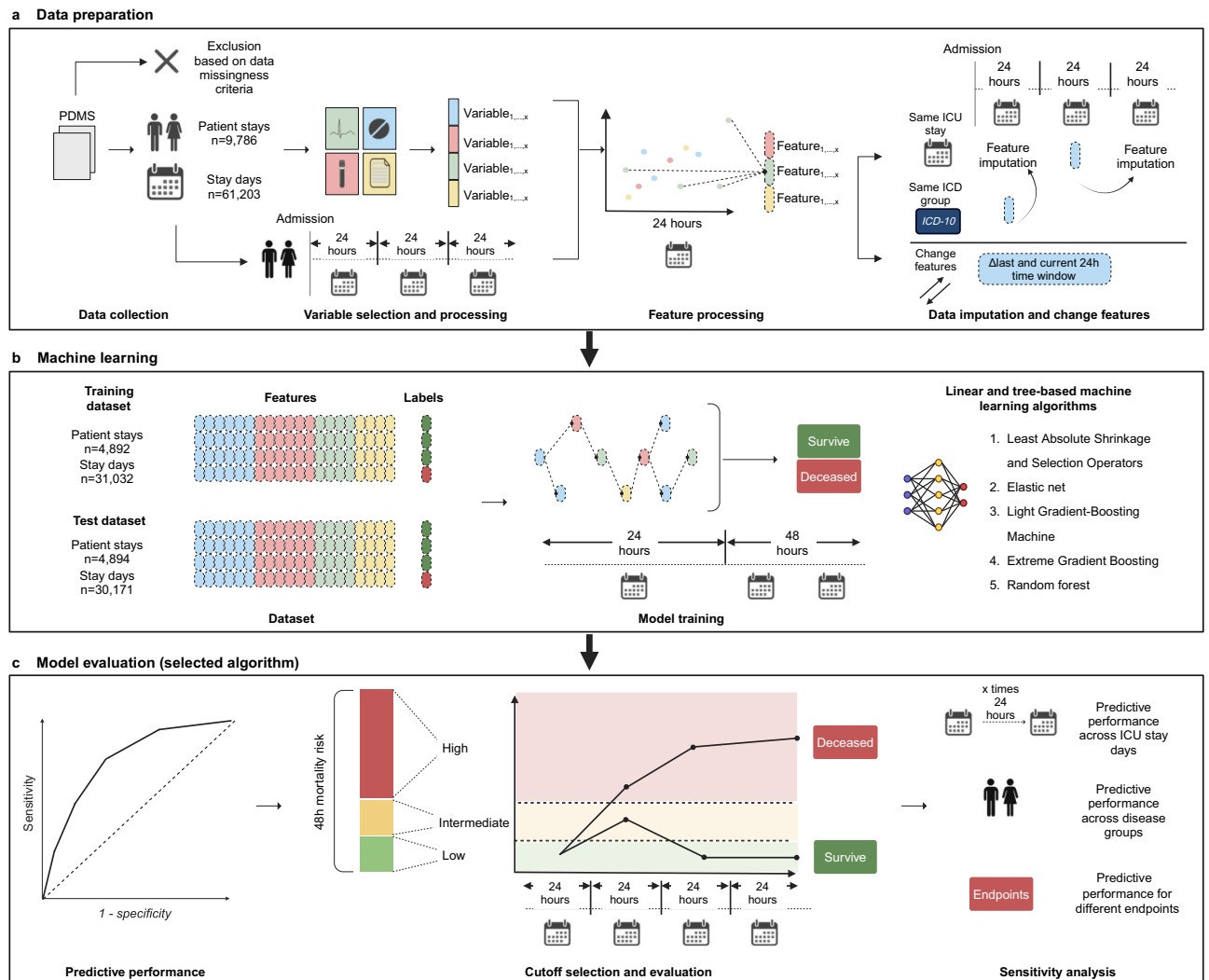

**Fig. 1 | Development of a machine learning algorithm to predict 48-hour mortality across ICU care. a** Data preparation: ICU patient data recorded in a patient data management system (PDMS) were exported from the Data Integration Center of the University Medical Center Mannheim, Germany. Data was excluded based on exclusion criteria. Variables included vital parameters, medications, laboratory results, treatments, and outcome metrics and were aggregated based on 24 h time intervals starting at admission. All available static and dynamic categorical and continuous aggregate variables were selected as features and underwent feature processing. Missing values were imputed using previous 24 h time intervals where available or median values from corresponding ICD-10 based disease groups. A separate dataset was created that contained change features indicating day-over day

change. **b** Machine learning: the final dataset was split into a training and test sets. Applying five different linear and tree-based machine learning algorithms, models were trained to predict 48 h ICU mortality at the end of each 24 h time interval. **c** Model evaluation: for the selected LGBM-48 algorithm, performance was assessed using area under the receiver operating characteristics curve. Selection of cutoffs was performed to identify and map low-, intermediate and high-risk categories based on predicted 48 h mortality risk across the ICU stay. Lastly, sensitivity analyses were performed for different stay days across the ICU stay, different disease groups and endpoints.

prediction horizon. This decision reflects a pragmatic compromise between clinical applicability and data availability. In routine ICU practice, many input variables, particularly laboratory parameters, are recorded at most once per day. More frequent prediction intervals (e.g., every 6 or 12 hours) would result in increased data sparsity and may require imputation, potentially reducing model robustness. Furthermore, a 48 h prediction window allows sufficient time for clinical teams to initiate interventions, while the 24 h update cycle aligns with daily clinical workflows and reduces alert burden on healthcare staff.

### Calculation of the SOFA and the SAPS II/TISS-10 score
To benchmark the performance of our machine learning algorithms against established standards, we calculated the SOFA score and SAPS II/TISS-10 score. The SAPS II/TISS-10 scores were automatically recorded daily and documented in the electronic patient file of the PDMS. When multiple SAPS

II/TISS-10 values existed for a stay day, the mean was used for aggregation. Among the 73,440 ICU stay days, 30.3% of SAPS II/TISS-10 values were missing, and these missing values were not imputed.

Conversely, the SOFA score was calculated retrospectively, with missing values handled according to the procedure outlined above, based on Vincent et al. 1996[9]. Comparisons involving SAPS II/TISS-10 were conducted on the subset of data where values were available. An overview of all variables used, including those for the ML algorithms, is shown in Supplementary Table 4.

### Outcome definitions
The primary outcome of our study was defined as 48 h mortality. For each patient stay, a stay day was labeled "deceased" if the ICU stay ended within 48 h after that day and resulted in the patient's death. Otherwise, the stay day was labeled "survived" (Fig. 1).

## Table 1 | Baseline characteristics of all included patients and stratified by training and test dataset

|  | All n = 9786 | Training dataset n = 4892 | Test dataset n = 4894 |
|---|---|---|---|
| Demographics and medical history |  |  |  |
| Age (years), median (IQR) | 66 (55–77) | 67 (55–76) | 66 (56–77) |
| Sex, No. (%) |  |  |  |
| Female | 3879 (39.6) | 1941 (39.7) | 1938 (39.6) |
| Male | 5907 (60.4) | 2951 (60.3) | 2956 (60.4) |
| BMI (kg/m²), median (IQR) | 26.1 (23.5–29.4) | 26.1 (23.5–29.4) | 26.1 (23.5–29.4) |
| Diabetes, No. (%) | 3300 (33.7) | 1657 (33.9) | 1643 (33.6) |
| Hypercholesterolemia, No. (%) | 936 (9.6) | 483 (9.9) | 453 (9.3) |
| Arterial hypertension, No. (%) | 3595 (36.7) | 1819 (37.2) | 1776 (36.3) |
| Chronic kidney disease (any stage), No. (%) | 2700 (27.6) | 1347 (27.5) | 1353 (27.6) |
| Renal replacement therapy, No. (%) | 1454 (14.9) | 694 (14.2) | 760 (15.5) |
| NIV/IMV, No. (%) | 4680 (47.8) | 2368 (48.4) | 2322 (47.4) |
| Main diagnosis[1], No. (%) |  |  |  |
| Diseases of the digestive system | 835 (8.5) | 414 (8.5) | 421 (8.6) |
| Diseases of the nervous system | 250 (2.6) | 129 (2.6) | 121 (2.5) |
| Injury, poisoning and certain other consequences of external causes | 1065 (10.9) | 529 (10.8) | 536 (11.0) |
| Certain infectious and parasitic diseases | 365 (3.7) | 173 (3.5) | 192 (3.9) |
| Diseases of the circulatory system | 3315 (33.9) | 1669 (34.1) | 1646 (33.6) |
| Neoplasms | 1630 (16.7) | 834 (17.0) | 796 (16.3) |
| Diseases of the respiratory system | 1235 (12.6) | 607 (12.4) | 628 (12.8) |
| Diseases of the genitourinary system | 270 (2.8) | 122 (2.5) | 148 (3.0) |
| Other | 821 (8.4) | 414 (8.5) | 406 (8.3) |
| Admission category, No. (%) |  |  |  |
| Medical | 4420 (45.2) | 2217 (45.3) | 2203 (45.0) |
| Scheduled surgery | 784 (8.0) | 385 (7.9) | 399 (8.2) |
| Unscheduled surgery | 3362 (34.4) | 1669 (34.1) | 1693 (34.6) |
| Other | 1220 (12.5) | 621 (12.7) | 599 (12.2) |
| Admission source, No. (%) |  |  |  |
| Emergency Room | 3031 (31.0) | 1546 (31.6) | 1485 (30.3) |
| Other ICU | 2247 (23.0) | 1123 (23.0) | 1124 (23.0) |
| Hospital ward | 3152 (32.2) | 1556 (31.8) | 1596 (32.6) |
| External hospital | 887 (9.1) | 431 (8.8) | 456 (9.3) |
| Other | 469 (4.8) | 236 (4.8) | 233 (4.8) |
| Length of ICU stay |  |  |  |
| 1 day, No. (%) | 3032 (31.0) | 1500 (30.7) | 1509 (30.8) |
| 2 days, No. (%) | 1589 (16.2) | 797 (16.3) | 792 (16.2) |
| ≥ 3 days, No. (%) | 5165 (52.8) | 2595 (53.0) | 2570 (52.5) |
| Length of ICU stay (days), median (IQR) | 3.3 (1.8–8.0) | 3.3 (1.8–8.0) | 3.2 (1.8–8.0) |
| Mortality, No. (%) |  |  |  |
| ICU mortality | 1528 (15.6) | 740 (15.1) | 788 (16.1) |

## Table 1 (continued) | Baseline characteristics of all included patients and stratified by training and test dataset

|  | All n = 9786 | Training dataset n = 4892 | Test dataset n = 4894 |
|---|---|---|---|
| In-hospital mortality | 2684 (27.4) | 1338 (27.4) | 1346 (27.5) |

*BMI* body mass index. *ICU* intensive care unit. *IQR* interquartile range. *IMV* invasive mechanical ventilation. *NIV* noninvasive ventilation.

Additional sensitivity analyses were conducted using a 24 and 72 h mortality end point that followed the same definition as the 48 h mortality. ICU mortality was determined using ICU stay data, with a "deceased" label applied if the patient died at the conclusion of the ICU stay. Hospital mortality was labeled "deceased" if the patient died at the end of the hospital stay. ICU mortality and hospital mortality were trained and tested using only the first day of each patient stay. For the development of the machine learning algorithms for the prediction of ICU and hospital mortality, the same variables were used as for 48 h mortality.

### Model development and calibration

Model development was conducted using R Statistical Software (v4.2.3)[21] and the tidymodels framework package (v1.0.0)[22]. An overview of the model development process is provided in Fig. 1.

First, the data were divided into equal-sized training and test datasets, stratified by ICD-10 diagnosis groups. All stay days for a given patient stay were assigned entirely to either the training dataset or the test dataset, including the difference features (Supplementary Table 5). Next, for both training datasets, different linear and tree-based algorithms were applied with the goal of developing the algorithm with the best predictive performance. Linear methods included lasso and elastic net[23]. Lasso is a form of logistic regression that incorporates a linear regularization term, which helps to prevent collinearity and effectively select features. Conversely, elastic net combines the Lasso with ridge regression, integrating a quadratic regularization term. Three tree-based algorithms were used (LightGBM[24], XGBoost[25], and random forests[26]). All these methods are ensemble-based. Random forests consist of a forest ensemble of decision trees, while LightGBM and XGBoost leverage gradient boosting to generate boosted decision trees.

All analyses of the training dataset were performed using tenfold cross-validation. This approach ensured sufficient representation of smaller diagnostic subgroups and avoided data sparsity in clinically relevant categories. While fully nested cross-validation would provide a more rigorous estimate, we opted for this simpler and more transparent strategy, consistent with clinical ML practice[27]. For hyperparameter tuning, a grid search was utilized for the Lasso method, which involves only one hyperparameter. Bayesian optimization was employed for the other algorithms (Tidy Tuning Tools in Kuhn M (2024)[28]). All algorithms were optimized for the Brier score without any over-sampling or under-sampling to ensure proper calibration[29].

Lastly, the calibration of the models was evaluated using a calibration curve, as described by Wilks (1990)[30]. The curve included 95% confidence bands. The need for recalibration was determined by whether the confidence band overlapped with the plot's diagonal line, representing the theoretical optimal calibration.

### Assessment of feature importance

To visualize the impact of the most influential features on one single prediction, we employed SHAP values[31], derived from Shapley values in game theory, treating each feature as a player and the prediction as the payout shared fairly among them. To identify the characteristics responsible for a patient's health deterioration, we analyzed the differences in SHAP scores to determine where changes in mortality risk originated. For clarity, the SHAP values of features representing aggregations of a single measured variable

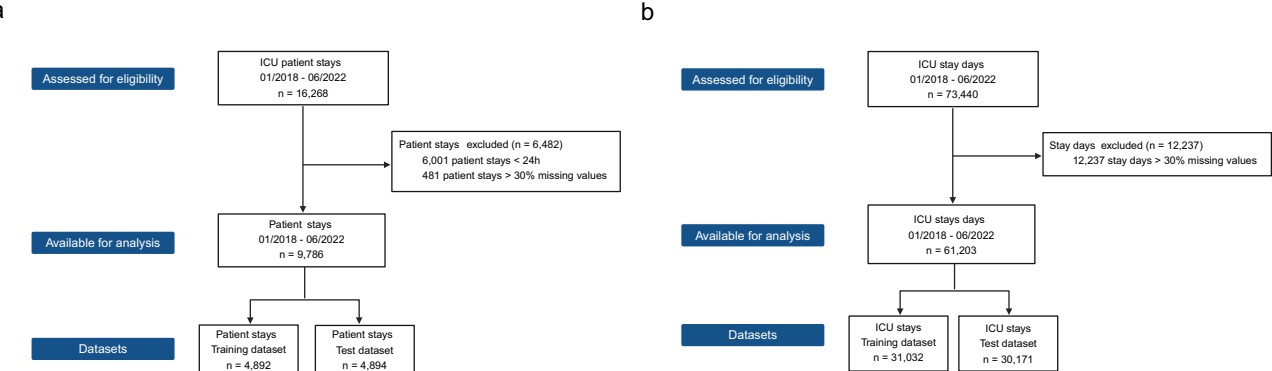

**Fig. 2 | Flow chart illustrating data inclusion based on patient stays and stay days. a** Flow chart for ICU patient stays. **b** Flow chart for ICU stay days (defined as 24 h intervals starting with admission).

were summed. Adding and subtracting SHAP values yielded consistent results due to their linear nature.

### Clinical threshold selection

To select clinical thresholds for predicted 48 h mortality risk, we calculated the Youden's index[32] and the F1-score[33] based on stay days. Maximum Youden's index (lower threshold) and F1-score (upper threshold) were used to define three risk categories (low, intermediate, and high). Youden's index combines sensitivity and specificity into a single measure and gives equal weight to false positive and false negative results, while the F1-score is defined as the weighted mean of the positive predictive value (PPV) and sensitivity. This combined approach of utilizing both methods prioritizes specific aspects of decision-making across two critical transition zones (low-to-intermediate and intermediate-to-high risk). For the transition from low to intermediate risk, the focus is on maximizing overall classification efficiency by balancing sensitivity and specificity, as represented by the Youden Index. This method aims to identify as many at-risk patients as possible while minimizing the number of false positives. In contrast, during the transition from intermediate to high risk, the emphasis shifts to accurately identifying high-risk patients by optimizing the trade-off between PPV and sensitivity, which is captured by the F1-score. This approach ensures that patients with a high predicted 48-hour mortality risk are flagged accurately while reducing unnecessary alarms. This dual-threshold strategy facilitates an effective and targeted approach to risk stratification.

To plot mortality based on the duration spent in the high-risk category, we established density curves using Kernel Density Estimation over the relative number of days spent in the high-risk category. This analysis was performed based on patient stays of patients who either survived or succumbed and spent at least one day in the high-risk category. Mortality calculation was adjusted by the corresponding total number of patient stays.

### External validation using MIMIC-IV database

To assess generalizability, we externally validated the LGBM-48 h model using the publicly available MIMIC-IV database (version 3.1)[34], comprising ICU data from Beth Israel Deaconess Medical Center. Data were imported via PostgreSQL and processed with the same R pipeline as the internal ICCA dataset, with minimal adjustments for format compatibility. Since MIMIC-IV lacks admission diagnoses, we imputed missing values on ICU day 1 using the cohort median rather than diagnosis-stratified medians. The pretrained LGBM-48 h model was applied without retraining to evaluate out-of-the-box performance.

### Statistical analysis and reproducibility

All statistical analysis was performed using R Statistical Software (v4.2.3)[21]. For evaluating model performance and conducting model comparisons, we computed area under the receiver operating characteristic curves (AUR-OCs). Confidence bands depicted in figure plots represent 95% confidence

intervals (CI) and were derived using 2000 bootstrap replicates. CIs for the AUROCs were obtained using the DeLong method[35]. To assess the predictive performance of the SOFA and SAPS II/TISS-10 scores, we employed standard logistic regression. To ensure reproducibility, we implemented a code-based analysis pipeline and used fixed random seeds for model training and evaluation. Although traditional experimental replicates were not applicable, reproducibility was supported through stratified train–test splitting, tenfold cross-validation, and external validation using an independent cohort (MIMIC-IV).

## Results

### Patient population selection and characteristics

The complete dataset consisted of 16,268 patient stays, covering the period from January 2018 to May 2022. (Fig. 2a). Among these, 6001 patient stays (36.9%) were excluded because the patient stay was less than 24 h. An additional 481 patient stays (3.0%) were excluded due to having more than 30% missing data within the first 24 h.

From initial 73,440 stay days, a total of 12,237 (16.7%) stay days were excluded because they had more than 30% missing data (Fig. 2b). The final cohort consisted of 9786 eligible patient stays and 61,203 stay days (Fig. 2a, b). Both, the patient stay and stay day based datasets were split equally in a training ($n = 4892$ and $n = 31,032$ respectively) and test ($n = 4894$ and $n = 30,171$) dataset.

Baseline characteristics of all datasets are displayed in Table 1. The median age at admission was 66 years (interquartile range 55–77), and 39.6% of patients were female. Overall ICU and in-hospital mortality were 15.6 and 27.4%, respectively. Across 61,203 ICU stay-days included in the final cohort, 4.5% were labeled as deceased based on the 48 h mortality definition, indicating a moderate class imbalance in the primary outcome. In comparison, overall ICU mortality across all patient stays was 15.6%, and hospital mortality reached 27.4%. Despite the imbalanced nature of the 48 h mortality outcome, model development was conducted without applying over- or undersampling techniques. This approach was chosen to preserve the true outcome distribution and ensure accurate probability calibration. Model optimization relied on the Brier score, a strictly proper scoring rule that is particularly suitable for evaluating probabilistic predictions in imbalanced clinical datasets[36–38].

### Development of a dynamic machine-learning-based prediction algorithm for near-term ICU mortality

Next, we developed a dynamic ML-based algorithm for near-term ICU mortality prediction, designed to predict ICU mortality risk for the subsequent 48 h at the end of each 24 h stay day. To this end, we compared five established ML algorithms (two linear and three tree-based algorithms) using data from the training dataset. Among the ML algorithms, the LightGBM algorithm (LGBM-48 h) achieved the highest AUROC of 0.909 (95% CI: 0.901–0.917) in the training dataset with tenfold cross-validation,

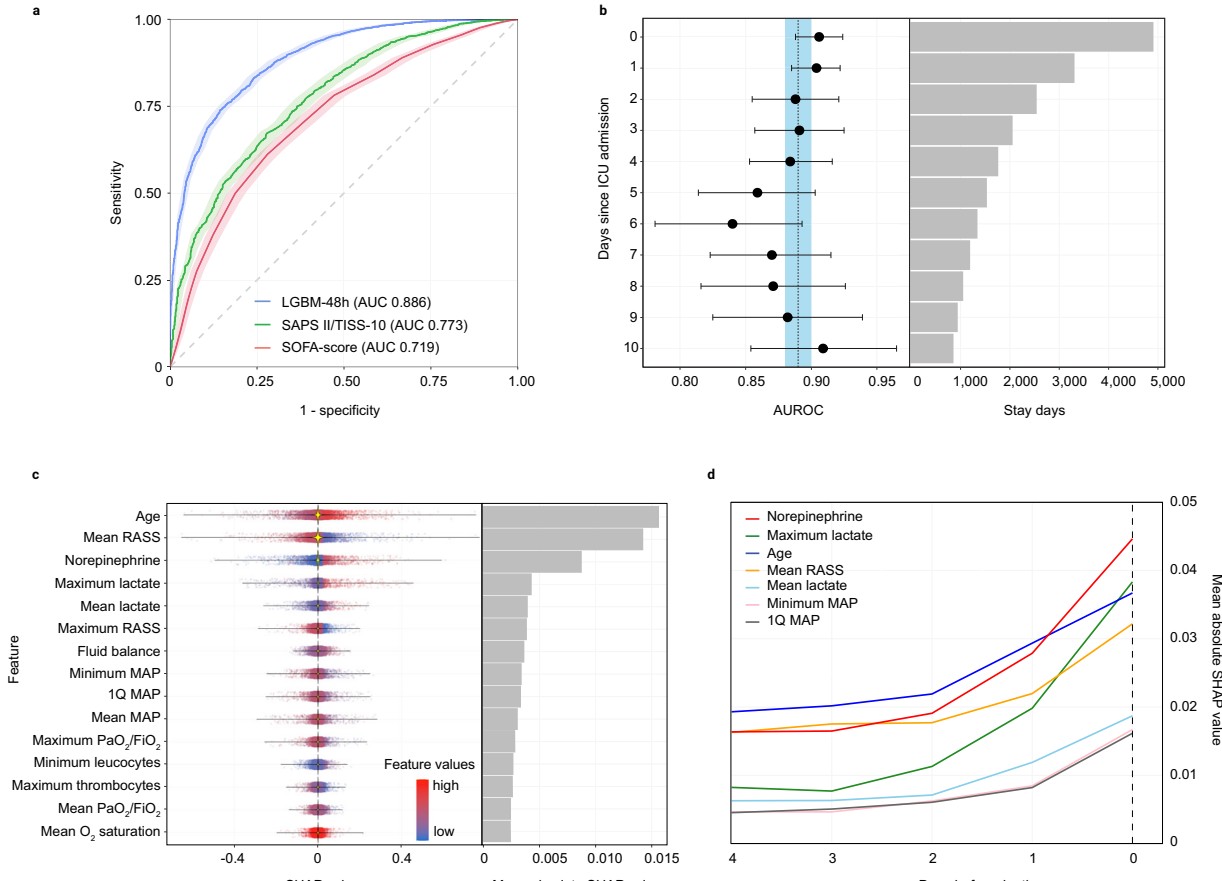

**Fig. 3 | Performance of the selected LGBM-48h model and feature importance.**
**a** Receiver operating characteristic curves illustrating the predictive performance for 48 h ICU mortality of the LGBM-48 h algorithm, the SAPS II/TISS-10 and the SOFA score. **b** Forest plot illustrating the predictive performance for 48 h ICU mortality of the LGBM-48 h algorithm, stratified by stay day after ICU admission. Black filled circles represent individual AUROC values for each ICU stay day after admission. Error bars indicate 95% confidence intervals derived using the DeLong method. Dotted line indicates area under the receiver operating characteristic curve (AUROC) for the LGBM-48 h algorithm, shaded blue area illustrate 95% confidence

intervals. Gray bars on the right indicate number of stay days that were included in the analysis ($n$ = 30,171 stay days in the test set). **c** Violin and dot plot illustrating the 15 features with the highest mean absolute SHAP values. The bar chart on the right summarizes the mean absolute SHAP value for each feature. **d** Line graph illustrating the change of mean absolute SHAP values for seven features with the highest absolute SHAP values at time of death. In patients that succumb, mean absolute SHAP values change in proximity to death.

and a comparable albeit slightly lower AUROC of 0.886 (95% CI: 0.878–0.895) in the test dataset (Supplementary Table 6) with an excellent calibration across the prediction range (Supplementary Fig. 1). Based on the AUROC, the LGBM-48 h algorithm was selected for further evaluation. Next, we compared the performance of the LGBM-48 h algorithm against the current clinical standard of care for mortality prediction on admission. In patients of whom all data was available to calculate SAPS II/TISS-10 ($n$ = 7783 stays in the test dataset), the LGBM-48 h algorithm showed a better AUROC of 0.886 (95% CI: 0.878–0.895) compared to SAPS II/TISS-10 with 0.773 (95% CI: 0.757–0.789) and the SOFA Score with 0.719 (95% CI: 0.701–0.736) (Fig. 3a).

Considering that the LGBM-48 h algorithm is intended to predict 48 h mortality for each day of an ICU stay, we subsequently examined whether its performance varied across the ICU stay. For this purpose, we assessed the algorithm's predictive performance over the first ten days of the ICU stay (Fig. 3b). AUROC for individual stay days, along with their respective CIs, did not differ significantly and indicated a stable predictive performance along the ICU stay.

To identify the features that contribute to the LGBM-48 h algorithm, a SHAP value analysis was performed. Supporting the dynamic prediction nature of the LGBM-48 h algorithm, features that reflect circulation (mean

and minimum arterial blood pressure [MAP], norepinephrine dose), tissue perfusion (mean and maximum lactate), vigilance (maximum and mean RASS), inflammation (maximum CRP, minimum leukocytes) as well as oxygenation and ventilation (mean $O_2$ saturation, mean and maximum $paO_2/F_iO_2$) thus indicate dynamic changes in patient status were identified as main contributors (Fig. 3c). Accordingly, only age (a static variable that does not change during ICU stay) was featured on top amongst all important features. Feature importance further changed with proximity to mortality in patients who succumbed (Fig. 3d). In this patient cohort, mean Richmond Agitation-Sedation Scale (RASS), lactate (both maximum and mean values), norepinephrine treatment, age and MAP increased in feature importance.

In summary, the LGBM-48h algorithm predicts 48 h ICU mortality with an excellent clinical performance along the ICU stay. Features that change dynamically during ICU stay contribute significantly to the LGBM-48 h algorithm performance and the algorithm outperforms the current clinical standard of care (SAPS II/TISS-10 and SOFA score).

## Selection of alarm thresholds for clinical application
The translation of predictive probabilities into meaningful clinical insights can be facilitated by the application of thresholds. Careful selection of

thresholds is important to avoid alert fatigue and increased work burden while ensuring that patients at heightened risk of mortality are accurately identified. In this study, threshold selection aimed to identify stay days associated with either a very low or a very high mortality risk. Stay days with a very low risk might identify patients who benefit from a shorter ICU stay, while those at intermediate or very high risk may require additional evaluation.

To this end, we performed a threshold analysis with the predicted 48 h ICU mortality risk generated by the LGBM-48 h algorithm for all stay days in the test dataset using Youden's index and the F1-score. To select thresholds, we used the maximum respective score (Supplementary Table 7). The threshold selected based on the maximum Youden's index corresponded to a predicted 48-hour mortality rate of 2.1% and led to a sensitivity of 85.7%, a specificity of 73.7%, and a total alarm rate of 29.0%, which corresponds to an average of 2.9 alarms per 24 h in a 10-bed ICU. In comparison, threshold selection based on the maximum F1-score corresponded to a predicted 48-hour mortality rate of 20.0% and resulted in a sensitivity of 43.7%, a specificity of 97.4%, and a total alarm rate of 4.5%, which translates to an average of 0.5 alarms per 24 h in a 10-bed ICU. Thus, maximizing Youden's index alone caused an unacceptably high alarm rate, whereas prioritizing the F-score alone led to an inappropriately low sensitivity for clinical practice. We therefore applied both derived cutoffs to categorize stay days into three risk groups (low, intermediate and high risk) based on predicted 48-hour mortality (Supplementary Table 8 and Fig. 4a). With this approach, prediction of either intermediate or high risk demonstrated a positive predictive value (PPV) of 13.7% and a negative predictive value (NPV) of 99.1%. Predicting high risk yielded a PPV of 45.2% and a NPV of 97.3% (Supplementary Table 8). Out of 30,171 stay days in the test dataset, 21,407 (71.0%) were classified as low risk, 7409 (24.6%) as intermediate risk and 1355 (4.5%) as high risk. To provide a more intuitive understanding of alarm patterns and risk dynamics, we visualized the average predicted mortality probability and corresponding alarm levels over time for both ICU survivors and non-survivors (Supplementary Fig. 2). Among non-survivors, a pronounced nonlinear increase in risk was observed prior to death, accompanied by frequent transitions between risk categories. In contrast, survivors showed relatively stable predicted probabilities with predominantly low-risk alarms throughout their ICU stay.

In summary, we derived thresholds that identify stay day categories with low, intermediate, and high risk for 48 h ICU mortality. Particularly the combined intermediate and high-risk stay day category yielded an excellent NPV, while the high-risk category provided an acceptable PPV.

## Change of predicted mortality risk longitudinally and identification of associated features through changes in SHAP values

Patient status may change dynamically during an ICU stay and may require adaptation of mortality risk estimates. Conceptually, ICU patients may either remain within their initial mortality risk category or experience changes in their mortality risk over the course of their stay.

In our study, 68.4% of patients consistently remained in their 48 h ICU mortality risk category (either low, intermediate or 48 h ICU high mortality risk) determined on admission. These categories effectively differentiated ICU mortality, with ICU mortality rates of 1.0% in the low-risk group, 32.6% in the intermediate-risk group, and 88.8% in the high-risk group (Supplementary Table 9). Conversely, 31.6% of patients experienced dynamic changes in their 48 h mortality risk, with 24.7% of patients changing between two and 6.9% between all three 48 h mortality risk categories. Additionally, ICU mortality differed in patients that experienced changes in 48 h risk category assignment compared to patients that remained within their 48-hour risk category determined on admission. Next, we assessed whether cumulative time in the high-risk category associated with higher ICU mortality. We observed that the fraction of stay days spent in the high-risk category per patients stay correlated with actual ICU mortality (Fig. 4b). This association was also evident for the low- and intermediate-risk groups, with the GAM-derived mortality curves demonstrating a continuous

increase in mortality for longer time spent in the intermediate-risk category, and a corresponding decrease for increasing exposure to the low-risk category (Supplementary Fig.3).

Interpreting the mortality risk of individual patients during their ICU stay relies heavily on the clinical context, particularly in recognizing changes in patient status and identifying the underlying factors driving these changes. Based on the predicted 48 h mortality, Fig. 4c depicts the longitudinal change of risk and risk categories in two patients who either survived their ICU stay or succumbed.

To identify features contributing to changes in a patient's mortality risk, we explored a method to visualize the variation in SHAP values across different stay days of a patient's stay. SHAP values of features that were aggregations of a single variable were combined. To compare SHAP values of different days, the corresponding SHAP values were subtracted, and because SHAP values are linear, adding and subtracting them produced SHAP-like results. This method allowed for the identification of changes in feature importance that contribute to the change of mortality risk as predicted by the LGBM-48h algorithm. As an example, SHAP values were derived for the third-to-last (>48 h), second-to-last (24–48 h), and last day (<24 h) in a selected patient who succumbed and was mapped as non-survivor in Fig. 4c. We then calculated the differences in SHAP values between the third-to-last and second-to-last days (Fig. 4d), as well as between the second-to-last and last days before death (Fig. 4e). In this patient's case, changes in importance of respiratory features (PEEP, $O_2$ saturation) preceded those in circulatory features (MAP, lactate) that contributed to the change in 48 h mortality risk predicting and ultimately indicated deterioration of the patient's clinical status.

In summary, while most patients show stable mortality risk throughout their ICU stay as assessed by the LGBM-48 h algorithm, a notable number of patients experience dynamic changes in their mortality risk. Specifically, actual mortality increases over the duration that patients remain in the predicted high-risk category.

## Predictive performance of the LGBM-48 h algorithm across different subgroups and endpoints

To provide a more detailed assessment of the LGBM-48 h algorithm, additional sensitivity analyses were conducted across various subgroups and different endpoints. To evaluate the algorithm's performance across different diseases, subgroups were defined based on admission diagnoses categorized by ICD-10 codes. The LGBM-48h algorithm demonstrated similar AUROC values for the ICD-10-based subgroups compared to the overall study population (Fig. 5a), with the exception of the subgroups "diseases of the genitourinary system" and "neoplasms" (where it performed better) and "diseases of the respiratory system" (where it performed worse). Similar results were observed when using ICD-10 codes representing the final diagnosis instead of the admission diagnosis, with the exception of the subgroup of "neoplasms" (where it performed better) and "diseases of the respiratory system" (where it performed worse) (Fig. 5b).

To provide flexibility in predicting both shorter and longer mortality time points, we evaluated the performance of the LGBM-48 h algorithm for predicting 24 and 72 h mortality. Compared to the prediction of 48 h mortality, the predictions for 24 and 72 h mortality exhibited marginally different AUROC values, with a gradual decline in predictive performance as the time window extended. Specifically, the AUROC was 0.905 (95% CI 0.895–0.916) for 24-hour mortality, 0.886 (95% CI 0.878–0.895) for 48 h mortality, and 0.873 (95% CI 0.866–0.882) for 72-hour mortality in the test dataset (Fig. 5c).

Finally, the LGBM-48 h algorithm was compared with other LGBM models for predicting ICU and hospital mortality. For these comparisons, only data from the first 24 h of ICU admission were utilized to develop the machine learning models for ICU and hospital mortality, employing the same variables as used in the LGBM-48h algorithm. The performance of the machine learning models for predicting ICU and hospital mortality demonstrated lower AUROC values than the LGBM-48h algorithm for predicting 48 h mortality (AUROC LGBM-48 h: 0.886, 95% CI 0.878–0.895;

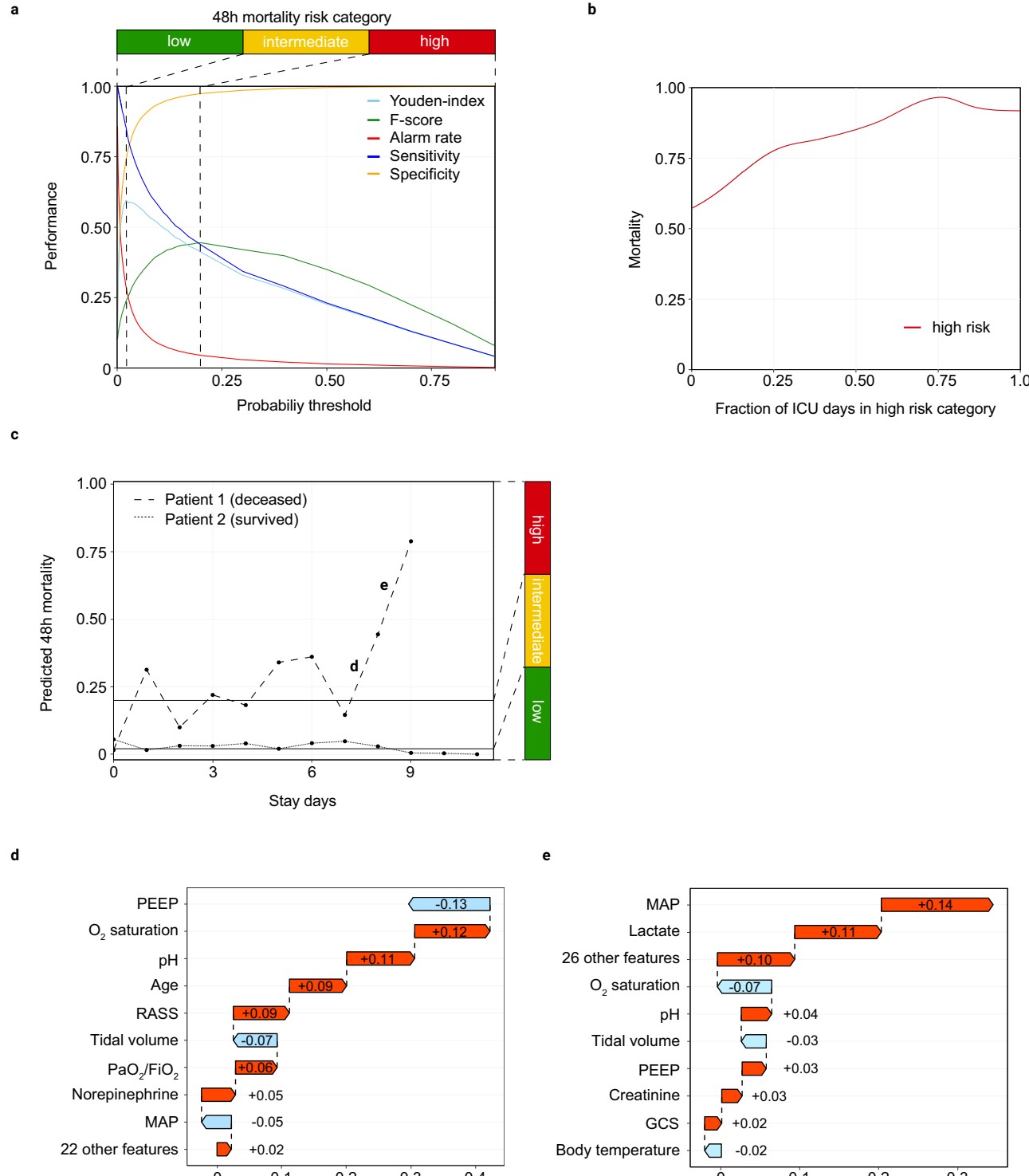

**Fig. 4 | Cut off selection to map 48 h mortality risk across the ICU stay and identification of features associated with change. a** Threshold plot showing Youden's index, F-score, alarm rate, sensitivity, and specificity to differentiate three risk categories for predicted 48 h mortality. The lower threshold was selected based on the maximum Youden's index, while the upper threshold was selected based on the maximum F-score. **b** Line plot illustrating ICU mortality based on fraction of ICU days in high-risk category per patient stay. Mortality was calculated from weighted density curves that were calculated by Kernel Density Estimation for patients with at least one day in the high-risk category. **c** Line plot depicting 48 h mortality risk calculated by the LGBM-48 algorithm for each day of the ICU stay of two patients (survivor, dashed line; non-survivor, dotted line). Indicators d and e refer to Figs. 4d and 4e that illustrate the change in SHAP values as 48 h mortality risk increases in the patient that succumbed. Horizontal solid lines identify lower and upper 48 h mortality risk thresholds that differentiate low, intermediate, and high risk. **d,e** calculated differences in SHAP values between the third-to-second (**d**) and second-to-last (**e**) 24 h interval before death for the non-surviving patient shown in Fig. 4c. The differences of SHAP value compared to SHAP values alone identifies shifts in feature importance affecting mortality risk predicted by the LGBM-48 h algorithm.

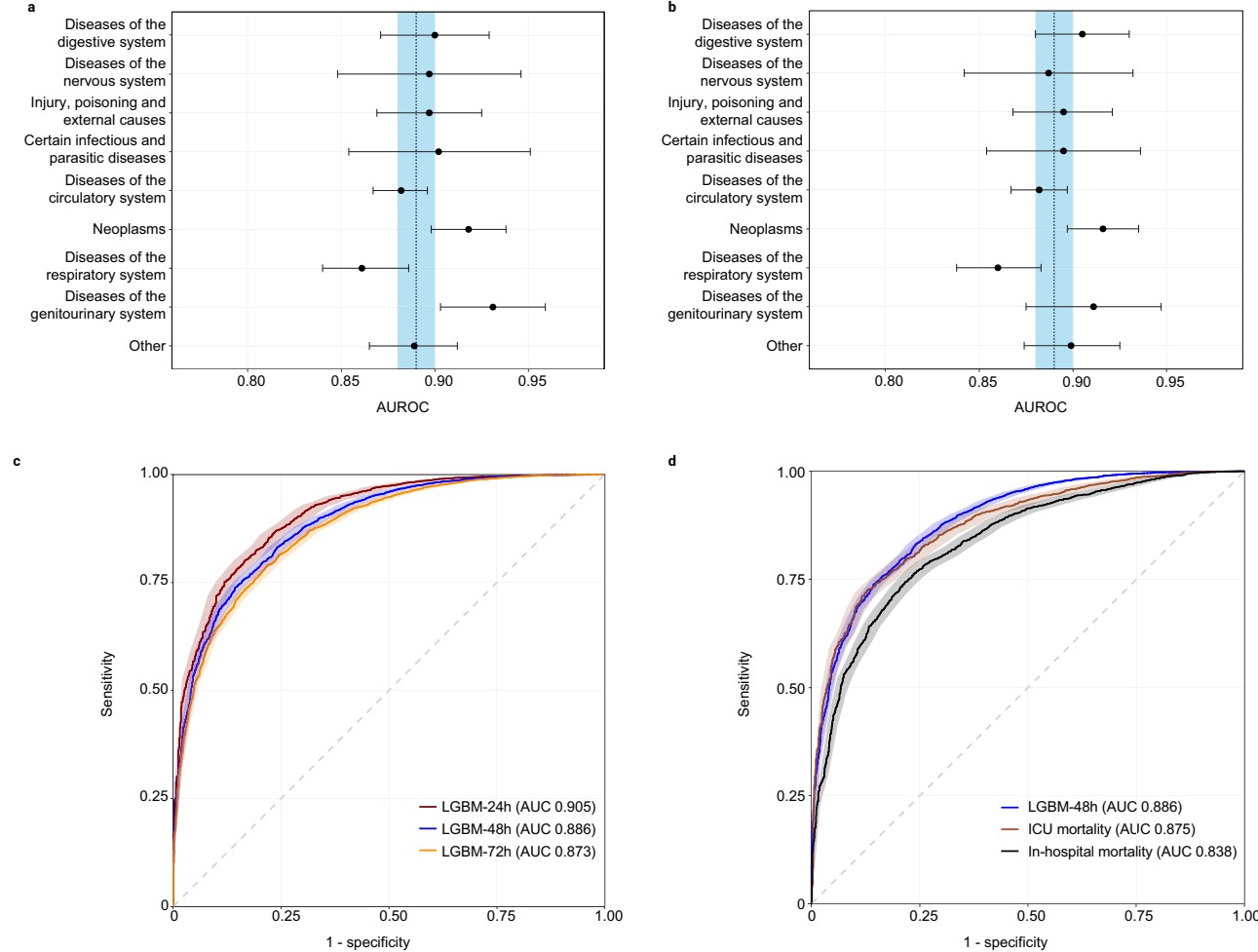

**Fig. 5 | Sensitivity analysis. a** Forest plot illustrating the predictive performance for 48 h ICU mortality of the LGBM-48h algorithm, stratified by ICD-10 diagnosis group assigned on admission. Black filled circles represent AUROC values for each admission diagnosis group based on ICD-10 categories. Error bars indicate 95% confidence intervals derived using the DeLong method. Dotted line indicates area under the receiver operating characteristic curve (AUROC) for the LGBM-48 h algorithm, shaded blue area illustrate 95% confidence intervals (*n* = 30,171 stay days in the test set). **b** Forest plot illustrating the predictive performance for 48-hour ICU mortality of the LGBM-48 h algorithm, stratified by ICD-10 diagnosis group assigned after review of all available patient data at the end of the respective ICU stay.

Black filled circles represent AUROC values for each final diagnosis group based on ICD-10 categories. Error bars indicate 95% confidence intervals derived using the DeLong method. Dotted line indicates area under the receiver operating characteristic curve (AUROC) for the LGBM-48h algorithm, shaded blue area illustrate 95% confidence intervals (*n* = 30,171 stay days in the test set). **c** Receiver operating characteristic curves comparing LGBM algorithms that predict 24 h ICU mortality (LGBM-24h), 48 h ICU mortality (LGBM-48h) and 72 h ICU mortality (LGBM-72 h), respectively. **d** Receiver operating characteristic curves comparing LGBM algorithms that predict 48 h ICU mortality (LGBM-48 h), overall ICU mortality and in-hospital mortality.

AUROC ICU mortality: 0.875, 95% CI 0.863–0.887; AUROC hospital mortality: 0.838, 95% CI 0.826–0.850) (Fig. 5d).

Our results indicate that the LGBM-48 h algorithm demonstrates consistent predictive performance across various underlying diseases within an ICU population. Adjusting the prediction time window to suit clinical needs also yields satisfactory performance, albeit with a slight decrease when the window is extended. Lastly, predicting common endpoints such as ICU and in-hospital mortality shows lower, yet still acceptable, predictive performance compared to predictions made within a defined time window.

**External validation in the MIMIC-IV cohort**
To evaluate the generalizability of the LGBM-48 h model, we applied the pretrained algorithm to the MIMIC-IV. A flow chart illustrating data inclusion and exclusion based on patient stays and stay days is shown in Supplementary Fig. 4. Detailed characteristics of the MIMIC-IV validation cohort are presented in Supplementary Table 11.

Overall, LGBM-48h achieved an AUROC of 0.859 (95% CI: 0.849–0.870) in the MIMIC-IV dataset, indicating good discriminative

performance, albeit lower than in the derivation cohort (AUROC 0.889, 95% CI: 0.878–0.895, Supplementary Fig. 5).

To assess the applicability of previously established thresholds (Youden's index 0.021 to differentiate low from intermediate risk, F1-score 0.200 for distinguish between intermediate and high risk), we evaluated their performance in the MIMIC-IV cohort. Among 39,259 ICU stay-days, 76.4% were classified as low risk, 20.3% as intermediate risk, and 3.3% as high risk, closely resembling the distribution in the internal dataset (70.9, 24.6, and 4.5%, respectively; see Supplementary Table 8). Using these predefined thresholds, the model yielded a sensitivity of 76.6%, specificity of 78.4%, negative predictive value (NPV) of 99.1%, and positive predictive value (PPV) of 10.0% for the low threshold, and a sensitivity of 34.6%, specificity of 97.7%, NPV of 97.9% and PPV of 32.0% for the high threshold.

We then derived dataset-specific thresholds based on the maximum Youden's index (0.018) and F1-score (0.190), as shown in Supplementary Table 12 and Supplementary Fig. 6. Compared to the thresholds derived from University Medical Center Mannheim ICU dataset (Youden´s index 0.021 and F1-score 0.200), the cutoffs based on the MIMIC-IV cohort showed subtle differences. At the lower threshold of 0.018, the model

achieved a sensitivity of 79.6%, specificity of 75.9%, NPV of 99.2%, and PPV of 9.4%. At the F1-based threshold of 0.190, sensitivity was 35.9%, specificity 97.5%, NPV 98.0%, and PPV 31.4%. These findings support the generalizability of the LGBM-48h algorithm across different ICU populations.

## Discussion

ICU care is characterized by high mortality rates and constrained resources, underscoring the need for accurate and adaptive mortality prediction that can be effectively translated into clinical decision-making and resource management. Given the dynamic nature of mortality risks, which may fluctuate with daily changes in a patients' condition, we developed the LGBM-48h algorithm to evaluate 48-hour mortality risk throughout the ICU stay.

The LGBM-48h algorithm developed in our study delivers excellent predictive performance throughout the ICU stay, surpassing the SAPS II/ TISS-10 and SOFA scores. By establishing thresholds based on predicted 48 h mortality risk, we classified individual predictions into low, intermediate, and high mortality risk categories. Beyond single-point predictions, extended duration in the high-risk category was closely associated with an increase in mortality risk. The analysis of longitudinal changes in SHAP values helped identify factors driving fluctuations in mortality risk. Finally, adjusting the prediction time window to suit clinical requirements also resulted in satisfactory predictive performance, albeit with a slight decline as the time window extended.

In this study, we deliberately chose to use LightGBM as a classical tree-based machine learning algorithm due to its high performance, low computational cost, and strong interpretability via SHAP values. These characteristics are particularly useful in clinical contexts where transparency and computational efficiency are critical. Our intention was to first explore how well established, traditional ML methods perform in the setting of dynamic ICU mortality prediction before transitioning to more complex deep learning approaches such as LSTM networks or Transformer-based models[39]. This stepwise approach allows for better benchmarking and ensures clinical applicability at each development stage. To incorporate temporal dynamics, we engineered difference features that quantify the change in key variables over time, enabling the model to partially account for trends and deterioration patterns without requiring sequential modeling. Although ensemble methods can theoretically improve performance, our best-performing models, including LightGBM, XGBoost, and Random Forest, are already ensemble-based approaches built on decision trees. Given their structural similarity, additional stacked ensembling is unlikely to yield substantial gains, while significantly increasing model complexity and limiting clinical interpretability[40]. Future work could explore combining structurally different models, such as deep learning or probabilistic approaches, to assess whether heterogeneous ensembles offer practical advantages.

Established methods for predicting mortality in the ICU depend on patient data collected upon admission, using scoring systems such as the SOFA score, SAPS-II, APACHE score, LODS, and OASIS to estimate ICU mortality. With advancements in machine learning, new tools with significantly improved predictive performance have been developed[41,42]. These tools boast excellent AUROCs, reaching up to 0.977, whether applied to specific patient subgroups or the broader ICU population[12,42]. Typically, ICU prediction tools focus on endpoints like ICU or in-hospital mortality, predicted at a single time point, usually upon admission or within the first 24 h. Despite advancements in methodology and performance, converting mortality predictions into clinical decisions that could enhance outcomes remains challenging.

In selecting variables for model development, we deliberately focused on features that are routinely available in most ICU settings and easily retrievable from standard electronic health records. This pragmatic approach was intended to ensure that the algorithm can be implemented across a wide range of clinical environments, without requiring access to highly specialized diagnostics or infrastructure. While the model includes more variables than traditional scores, such as SOFA, this trade-off enables

substantially higher predictive performance and dynamic risk assessment throughout the ICU stay.

To address this issue, several studies have investigated dynamic models that continuously update and refine predictions over time using data generated throughout the ICU stay[15,19,43,44]. However, some of these models have demonstrated varying predictive performance during the ICU stay[15,19,43]. In our approach, we developed a model that updates every 24 h, incorporating both static and dynamic features, as well as changes in dynamic features over time, to predict 48 h mortality. The overall predictive performance of our model aligns with previous publications that predicted short-term ICU mortality[19,44]. Moreover, the predictive performance of our model remained stable throughout the first 10 days of ICU stay and demonstrates applicability throughout the ICU stay and across a multitude of underlying diseases.

To address the uncertainty regarding the time-to-endpoint when predicting overall ICU mortality, some published algorithms focus on predicting short-term rather than overall ICU mortality[19,44]. The rationale behind this approach is to promptly identify either patients at imminent risk (enabling diagnostic or therapeutic interventions that could potentially alter the patient's trajectory) or those with a very low mortality risk (potentially supporting triage or discharge decisions)[19]. To support this rationale, we derived a risk category-based framework to generate clinically meaningful insights and to facilitate translation into clinical practice. We demonstrate that using individual 24 h intervals of a patient's stay, 48-hour mortality prediction achieves an excellent NPV of over 99% when applying a threshold that distinguishes the low-risk from the intermediate and high-risk category. A threshold that differentiates high risk from intermediate and low risk is characterized by higher specificity and moderate PPV for 48-hour mortality. In addition, these categories are associated with ICU mortality in patients who remain in the determined categories on admission, while changes in 48 h risk categories modify ICU mortality rates. In addition, the relatively low positive predictive value (PPV) observed in our model reflects the inherent trade-off between sensitivity and specificity when predicting rare events, such as 48 h ICU mortality. In this study, we intentionally favored high sensitivity to minimize the risk of missing critical deterioration, accepting a lower PPV as a consequence. While this increases the number of false positives, many of these flagged patients may still have been clinically unstable and thus benefited from increased monitoring. Alarm fatigue is a well-documented phenomenon in intensive care settings, where frequent non-actionable alerts can desensitize staff, delay responses, and jeopardize patient safety[45]. To mitigate this, our model supports customizable thresholds, allowing hospitals to calibrate sensitivity and PPV based on clinical context, institutional workflow, and risk tolerance. Moreover, the optimal threshold should reflect the specific clinical goal whether early warning, triage support, or case prioritization and ideally involves clinical judgment alongside algorithmic predictions. Furthermore, future studies should investigate the downstream effects of model-generated alerts on clinical decisions and outcomes. In summary, this framework allows to monitor 48 h mortality risk across an ICU stay and to support treatment decisions based on the risk category and trajectory of change in 48 h mortality risk. Although we have not derived cutoffs for other time intervals, the overall predictive performance suggests that our approach can also be applied to shorter (i.e., 24 h) and longer (i.e., 72 h) mortality prediction periods, though with lower performance as the time window extends.

Previous studies have explored the concept of serial assessments to predict the performance of established mortality prediction scores[46,47]. For instance, trends in the SOFA score during the first 96 h were associated with varying mortality rates: higher (increasing SOFA score), comparatively lower (unchanged SOFA score), and lowest (decreasing SOFA score)[46]. In line with these findings, we demonstrate that patients with accumulating days in the high-risk category experienced an increasing mortality risk. This data supports the idea that integrating mortality predictions throughout the patient's stay enhances predictive performance, and monitoring mortality risk categories over time may facilitate this approach.

Interpretability, defined as the extent to which a user can understand the reason behind a model's decision, is crucial for clinical adoption. Previous research has used SHAP values to identify and interpret features contributing to ICU mortality[48]. Additionally, SHAP values have been tracked throughout the ICU stay to monitor feature importance as patients approach a defined endpoint[18]. In this study, we extend this concept by applying the SHAP method to highlight features driving longitudinal changes in mortality risk. This level of interpretability could support clinicians in identifying the underlying cause determining a patient's status, which is vital for timely clinical decision-making to personalize care and potentially alter outcomes.

The application of the pretrained LGBM-48h model to the MIMIC-IV dataset provides robust evidence for its generalizability. Without retraining or recalibration, the algorithm achieved high discriminatory performance. This suggests that the model captures fundamental physiological deterioration patterns that are resilient to variation in healthcare systems, documentation standards, and patient populations.

Importantly, both predefined alarm thresholds and thresholds optimized on the MIMIC-IV dataset resulted in comparable performance metrics. Minor deviations in *negative predictive value* and positive predictive value were expected due to differences in outcome prevalence and measurement frequency across cohorts[49].

These findings underscore that the LGBM-48h model is not overfit to a single institution or data source and support its applicability in diverse clinical settings. Nevertheless, local threshold adaptation may still enhance clinical utility depending on context-specific prevalence and decision thresholds.

While the LGBM-48h model demonstrated excellent discrimination and calibration across multiple settings, it is important to emphasize that predictive accuracy does not necessarily translate into clinical benefit. Risk prediction is only the first step in a broader chain of clinical decision-making. For a true improvement in patient outcomes, such as reduced ICU mortality or improved resource allocation, further steps are required, including seamless integration into clinical workflows, real-time availability, and appropriate clinical response to predictions. Future prospective implementation studies are needed to determine whether the use of dynamic mortality risk predictions can directly influence treatment strategies and improve outcomes in ICU settings.

Despite these promising results, several limitations of our study should be acknowledged: First, the LGBM-48 h algorithm was trained and tested at a single hospital. Although the data was collected from patients across multiple ICUs and disease groups, external datasets are needed to validate our findings. Second, our study utilized a retrospective design. Prospective implementation studies are necessary to demonstrate the feasibility and clinical utility of our algorithm. Third, our stringent exclusion criteria led to the omission of more than one-third of all patient stays and 16.7% of all stay days. Additional data is required to assess the performance in a more inclusive patient population. Lastly, variable selection was based on clinical expertise and did not include interventions. Incorporating feature selection methodologies and including interventions may further enhance the performance of our model.

Finally, although our model shows strong predictive performance, the impact on clinical outcomes such as reduced mortality or improved care processes has not yet been demonstrated. The present study does not evaluate how risk predictions influence clinical decisions or patient trajectories. Therefore, future prospective implementation studies are required to assess the real-world utility and effectiveness of the LGBM-48 h algorithm in improving ICU outcomes.

## Conclusion

In conclusion, the LGBM-48h algorithm provides a dynamic, clinically applicable framework for 48-hour ICU mortality risk prediction. By establishing mortality risk-based categories and enhancing interpretability through the provision of longitudinal changes in SHAP values, the framework may help to identify changes in clinical status that affect outcomes and support care decisions. Further real-time and external validation are needed to establish the LGBM-48h algorithm as a tool for supporting clinical decision-making and resource management in ICU settings.

## Data availability

The datasets generated and/or analyzed during the current study are not publicly available. Data will be made available upon reasonable request to the corresponding author, subject to approval by the relevant institutional committees and in accordance with all applicable laws and regulations. Requests will be processed within 6–8 weeks. Data usage will be governed by a data use agreement that prohibits re-identification and limits usage to the approved scientific purpose. Source data for all main figures are provided as Supplementary Data in Excel format.

## Code availability

The code used for data preprocessing, model training and evaluation will be made available upon reasonable request for academic, non-commercial use. Requests should be directed to the corresponding author and will be subject to approval by the relevant institutional committees and in accordance with all applicable laws and regulations. All code was written using R Statistical Software (v4.2.3)[21] and the tidymodels framework package (v1.0.0)[22].

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

## Acknowledgements

This work received no external funding. Figures 1 and 2 were created with BioRender.com.

## Author contributions

Conceptualization: S.B., M.B., S.L., and T.B.; data curation and formal analysis: S.B., M.B., and T.B.; validation: S.B., M.B., S.L., L.H., V.S.-L., T.H., M.T., D.D., and T.B.; visualization: S.B., M.B., and T.B.; writing – original draft: S.B., M.B., and T.B.; writing – review and editing: All authors. All authors have given final approval of the version to be published and agree to be accountable for all aspects of the work.

## Funding

## Competing interests

T.B. is an employee and shareholder of Roche Diagnostics International, Rotkreuz, Switzerland. All other authors declare that they do not have a conflict of interest.

## Additional information

[1]Cardiology, Angiology, Haemostaseology, and Medical Intensive Care, Medical Centre Mannheim, Medical Faculty Mannheim, Heidelberg University, Heidelberg, Germany. [2]European Centre for AngioScience (ECAS), German Centre for Cardiovascular Research (DZHK) partner site Heidelberg/Mannheim, and Centre for Cardiovascular Acute Medicine Mannheim (ZKAM), Medical Centre Mannheim, Medical Faculty Mannheim, Heidelberg University, Heidelberg, Germany. [3]HMS Analytical Software GmbH, Heidelberg, Germany. [4]Department of Radiation Oncology, Medical Center Mannheim, Medical Faculty Mannheim, Heidelberg University, Heidelberg, Germany. [5]DKFZ Hector Cancer Institute, Medical Centre Mannheim, Medical Faculty Mannheim, Heidelberg University, Heidelberg, Germany. [6]Mannheim Institute for Intelligent Systems in Medicine (MIISM), Heidelberg University, Heidelberg, Germany. [7]Department of Anesthesiology and Surgical Intensive Care Medicine, Medical Center Mannheim, Medical Faculty Mannheim, Heidelberg University, Heidelberg, Germany. [8]Mannheim Institute for Innate Immunoscience (MI3), Medical Faculty Mannheim, Heidelberg University, Heidelberg, Germany. [9]These authors contributed equally: Simone Britsch, Markward Britsch. ✉e-mail: Simone.Britsch@umm.de

