## [Transparent Peer Review file · Communications Medicine]

An interpretable machine learning algorithm enables dynamic 48-hour mortality prediction during an ICU stay.

Corresponding Author: Dr Simone Britsch

Version 0:

Reviewer comments:

Reviewer #1

(Remarks to the Author)

This study showed that The LGBM-48h algorithm offers a dynamic framework for short-term ICU mortality prediction, necessitating further validation through real-time and external studies. This is generally a well written article with sophisticated techniques. I have a few comments:

1. The study lacks external validation, it would be better for a model to be tested in very different conditions, so as to avoid out-of-distribution bias.
2. there are many time-series models such as LSTM in ML, why do you think lightgbm is the first choice and this model may not well capture the temporal pattern of the model.
3. The outcome variable is imbalanced data. what is the proportion of deceased ones during the time frame?
4. It seems the SOFA score used much less variables for the parsimony principle, SOFA seems a better choice for practical use.
5. will the stacked ensemble model further improve the model performance? Since the authors developed several ML models, did you try to use ensemble to further improve the model performance (<https://doi.org/10.1016/j.lers.2021.12.003>)? There has been many empirical evidence that such ensemble models can help to achieve better accuracy than any individual model alone. At least you need to mention this potential.
6. It can be discussed further that while the model accuracy is important, but it may not translate to the clinical benefits such as reducing mortality in ICU. Thus, more work can be done to test the effectiveness of the model.

Reviewer #2

(Remarks to the Author)

Thank you for submitting your manuscript to Communications Medicine. I have the following comments:

1. Why did the authors design the experiment to update every 24 hours to predict 48-hour mortality, rather than making predictions every hour, or every 6 or 12 hours, to predict 12-hour or 24-hour mortality? The authors should provide a clear rationale for this design choice.
2. Please explicitly describe the inclusion and exclusion criteria for patient selection in the Methods section.
3. In line 206, the authors mention that the training dataset was used for tenfold cross-validation. Does this mean that tenfold cross-validation was only used for hyperparameter tuning? Why was the data split into equal-sized training and test datasets? Why not use tenfold cross-validation to partition training and test sets?
4. I strongly agree with the authors' effort to stratify patient risk, as prediction is not the ultimate goal. The analysis regarding threshold selection is interesting. Although Supplemental Table 9 shows the distribution of risk group categories for patient stays, I find the presentation somewhat unintuitive. Could the authors consider visualizing how the model-triggered alarms were distributed during ICU stays for test patients, similar to Figure 6 in the article "Continuous prediction and clinical alarm management of late-onset sepsis in preterm infants using vital signs from a patient monitor"? This would provide a more intuitive understanding of alarm patterns.
5. In lines 323–327, the authors mention that the PPV remains low, indicating many false alarms. I would like to know the

authors' thoughts on how to interpret or handle these false alarms.

6. In Figure 4b, what about the results for the intermediate and low-risk groups? Also, in Figure 4c, should patient 1 be labeled as deceased and patient 2 as survived?

7. I suggest the authors include the AUC values directly in ROC plots such as Figures 3a, 5c, and 5d. This would make it easier for readers to view the AUC results without referring back to the main text.

8. The authors included a summary at the end of almost every subsection in the Results. Please note that the Results section should objectively present experimental findings without summarizing or interpreting them. Any interpretations or summaries should be moved to the Discussion section.

Version 1:

Reviewer comments:

Reviewer #1

(Remarks to the Author)

My previous comments are well addressed.

Reviewer #2

(Remarks to the Author)

The author has addressed all of my comments, and I have no further comments.

Rebuttal Letter

The authors would like to express their sincere gratitude to the reviewers for their diligent review of our manuscript and for providing comprehensive and detailed feedback. We highly value the comments received, which we will address in detail below.

Reviewer 1:

This study showed that The LGBM-48h algorithm offers a dynamic framework for short-term ICU mortality prediction, necessitating further validation through real-time and external studies. This is generally a well written article with sophisticated techniques.

We would like to thank the reviewer for the positive assessment of our manuscript and the useful suggestions. Please find our detailed response to the observation and comments below, addressed point by point.

1. The study lacks external validation, it would be better for a model to be tested in very different conditions, so as to avoid out-of-distribution bias.

To evaluate the generalizability of our algorithm, we performed an external validation of the LGBM-48h model using the MIMIC-IV database, a publicly accessible ICU dataset from Beth Israel Deaconess Medical Center in the United States (PMID: 36596836). In this dataset, the LGBM-48h algorithm was implemented without any retraining to examine its performance in a different healthcare setting, thereby directly addressing potential issues related to out-of-distribution bias.

In this external validation, the LGBM-48h algorithm achieved an **AUROC of 0.859** (95% CI: 0.849–0.870). Albeit lower compared to the derivation dataset (AUROC 0.889, 95% CI: 0.878–0.895), the results indicate a robust discrimination despite differences between datasets.

To further assess generalizability, we validated the predefined cutoffs. These cutoffs yielded comparable values for the low and high threshold regarding sensitivity, specificity, PPV and NPV. As an example, specificity was 0.737 for the low threshold in the University Medical Center Mannheim ICU dataset, compared to 0.784 in the MIMIC-IV dataset.

In addition, we also determined new optimal cutoffs for the MIMIC-IV dataset using the Youden's index (0.018) and F1-score (0.190). These cutoffs were also in general comparable to those derived in the University Medical Center Mannheim ICU dataset (Youden's index 0.021, F1-score 0.200). Although minor differences were noticeable, the data suggest that the LGBM-48h algorithm is applicable in different settings without the need for recalibration.

A description of the external validation procedure has been added to the **abstract**, **methods section**, the **results** and **discussion section** has been updated accordingly. Additionally, we added five supplementary information:

Supplemental Figure 4 - Flow chart illustrating data inclusion from the MIMIC-IV dataset based on patient stays and stay days.

Supplemental Figure 5 - External validation of the LGBM-48h prediction model using the MIMIC-IV dataset.

Supplemental Figure 6 - Cutoff selection for 48-hour mortality risk across the ICU stay in the MIMIC-IV dataset.

Supplemental Table 8 - Predictive performance at the low and high threshold comparing the University Medical Center Mannheim ICU dataset and the MIMIC-IV dataset.

Supplemental Table 11 - Baseline characteristics of all MIMIC-IV patients used for external validation.

Supplemental Table 12 lists the optimal cutoff values derived from the MIMIC-IV cohort.

Abstract [page 2, lines 44-45]:

„External validation using the MIMIC-IV database yielded an AUROC of 0.859 (95% CI: 0.849–0.870).“

Methods [pages 10, lines 267-274]:

„3.11. External validation using MIMIC-IV database

To assess generalizability, we externally validated the LGBM-48h model using the publicly available MIMIC-IV database (version 3.1)³⁴, comprising ICU data from Beth Israel Deaconess Medical Center. Data were imported via PostgreSQL and processed with the same R pipeline as the internal ICCA dataset, with minimal adjustments for format compatibility. Since MIMIC-IV lacks admission diagnoses, we imputed missing values on ICU day 1 using the cohort median rather than diagnosis-stratified medians. The pretrained LGBM-48h model was applied without retraining to evaluate out-of-the-box performance.”

Results [pages 16-17, lines 452-476]:

„4.6. External validation in the MIMIC-IV cohort

*To evaluate the generalizability of the LGBM-48h model, we applied the pretrained algorithm to the MIMIC-IV. A flow chart illustrating data inclusion and exclusion based on patient stays and stay days is shown in **Supplemental Figure 4**. Detailed characteristics of the MIMIC-IV validation cohort are presented in **Supplemental Table 11**.*

*Overall, LGBM-48h achieved an AUROC of 0.859 (95% CI: 0.849–0.870) in the MIMIC-IV dataset, indicating good discriminative performance, albeit lower than in the derivation cohort (AUROC 0.889, 95% CI: 0.878–0.895, **Supplemental Figure 5**).*

*To assess the applicability of previously established thresholds (Youden’s index 0.021 to differentiate low from intermediate risk, F1-score 0.200 for distinguish between intermediate and high risk), we evaluated their performance in the MIMIC-IV cohort. Among 39,259 ICU stay-days, 76.4% were classified as low risk, 20.3% as intermediate risk, and 3.3% as high risk, closely resembling the distribution in the internal dataset (70.9%, 24.6%, and 4.5%, respectively; see **Supplemental Table 8**). Using these predefined thresholds, the model yielded a sensitivity of 76.6%, specificity of 78.4%, negative predictive value (NPV) of 99.1%, and positive predictive value (PPV) of 10.0% for the low threshold, and a sensitivity of 34.6%, specificity of 97.7%, NPV of 97.9% and PPV of 32.0% for the high threshold.*

*We then derived dataset-specific thresholds based on the maximum Youden’s index (0.018) and F1-score (0.190), as shown in **Supplemental Table 12** and **Supplemental Figure 6**. Compared to the thresholds derived from University Medical Center Mannheim ICU dataset (Youden’s index 0.021 and F1-score 0.200), the cutoffs based on the MIMIC-IV cohort showed subtle differences. At the lower threshold of 0.018, the model achieved a sensitivity of 79.6%, specificity of 75.9%, NPV of 99.2%, and PPV of 9.4%. At the F1-based threshold of 0.190, sensitivity was 35.9%, specificity 97.5%, NPV 98.0%, and PPV 31.4%. These findings support the generalizability of the LGBM-48h algorithm across different ICU populations.“*

Discussion [page 21, lines 578-589]:

„The application of the pretrained LGBM-48h model to the MIMIC-IV dataset provides robust evidence for its generalizability. Without retraining or recalibration, the algorithm achieved high discriminatory performance. This suggests that the model captures fundamental physiological deterioration patterns that are resilient to variation in healthcare systems, documentation standards, and patient populations.

Importantly, both predefined alarm thresholds and thresholds optimized on the MIMIC-IV dataset resulted in comparable performance metrics. Minor deviations in negative predictive value and positive predictive value were expected due to differences in outcome prevalence and measurement frequency across cohorts⁴⁹.

These findings underscore that the LGBM-48h model is not overfitted to a single institution or data source and supports its applicability in diverse clinical settings. Nevertheless, local threshold adaptation may still enhance clinical utility depending on context-specific prevalence and decision thresholds.“

2. There are many time-series models such as LSTM in ML, why do you think lightgbm is the first choice and this model may not well capture the temporal pattern of the model.

We intentionally selected LightGBM as our primary modeling approach for several key reasons. LightGBM provides an excellent balance of high predictive accuracy, low computational requirements, robustness to missing data, and—critically—model interpretability through SHAP values, which we believe is vital for clinical implementation.

Our initial focus was to evaluate well-established, classical machine learning models that are computationally efficient and less complex before progressing to deep learning architectures. This stepwise strategy facilitates rigorous benchmarking and clinical validation in a transparent and resource-conscious way.

To partially capture temporal dynamics, we engineered difference features that reflect day-to-day changes in each variable. These features enable the model to recognize trends and patterns of deterioration without relying on explicit time-series models such as long short-term memory (LSTM) networks.

We fully acknowledge that recurrent neural networks or transformer-based models might better capture complex temporal dependencies. We intend to investigate these advanced architectures in future studies, especially since their data preprocessing requirements differ substantially from those used here. We have included a corresponding statement in the revised Discussion section.

We have thus updated the **introduction** and **discussion** section to provide more clarity.

Introduction [page 3, lines 74-79]:

„Machine learning models for ICU prediction range from tree-based methods such as LightGBM to deep learning architectures like long short-term memory (LSTM) networks. While LSTM networks are capable of modelling temporal dependencies, they are often less interpretable and require higher computational resources. In contrast, gradient boosting tree algorithms offer strong predictive performance with lower complexity and greater transparency, which are important prerequisites for real-world clinical implementation¹⁴.“

Discussion [page 18, lines 491-500]:

„In this study, we deliberately chose to use LightGBM as a classical tree-based machine learning algorithm due to its high performance, low computational cost, and strong interpretability via SHAP values. These characteristics are particularly useful in clinical contexts where transparency and computational efficiency are critical. Our intention was to first explore how well established, traditional ML methods perform in the setting of dynamic ICU mortality prediction before transitioning to more complex deep learning approaches such as LSTM networks or Transformer-based models³⁹. This stepwise approach allows for better benchmarking and ensures clinical applicability at each development stage. To incorporate temporal dynamics, we engineered difference features that quantify the change in key variables over time, enabling the model to partially account for trends and deterioration patterns without requiring sequential modeling.“

3. The outcome variable is imbalanced data. what is the proportion of deceased ones during the time frame?

We thank the reviewer for drawing attention to this point. In our cohort, the overall ICU mortality was 15.6%, while 48-hour mortality, used for model training, occurred in 4.5% of the 61,203 patient stay days.

This information has now been incorporated into the **results** section of the revised manuscript. Additionally, we optimized our model using the Brier score without applying class rebalancing, to ensure proper calibration. We believe this approach is more clinically meaningful than training on artificially rebalanced data.

Results [page 11, lines 295-303]:

„Across 61,203 ICU stay-days included in the final cohort, 4.5% were labelled as deceased based on the 48-hour mortality definition, indicating a moderate class imbalance in the primary outcome. In comparison, overall ICU mortality across all patient stays was 15.6%, and hospital mortality reached 27.4%. Despite the imbalanced nature of the 48-hour mortality outcome, model development was conducted without applying over- or under-sampling techniques. This approach was chosen to preserve the true outcome distribution and ensure accurate probability calibration. Model optimization relied on the Brier score, a strictly proper scoring rule that is particularly suitable for evaluating probabilistic predictions in imbalanced clinical datasets^{36,37,38}.“

4. It seems the SOFA score used much less variables for the parsimony principle, SOFA seems a better choice for practical use.

We agree with the reviewer that the SOFA score's reliance on a small set of variables contributes to its simplicity and ease of use. However, our study shows that the LGBM-48h model significantly outperforms SOFA in predictive accuracy (AUROC 0.886 vs. 0.719), as detailed in the Results section. While ease of use is an important consideration for prognostic tools, predictive accuracy is equally critical and may lead clinicians to prefer the LGBM-48h algorithm.

To ensure the model remains user-friendly and clinically applicable, we carefully selected input features that are routinely collected in most ICU settings and are typically available in modern electronic health record (EHR) systems. This deliberate choice enhances the model's practical applicability across a wide range of hospitals, including those without highly specialized infrastructure. The model includes 131 features (28 raw variables), encompassing core vital signs, common laboratory parameters, and basic clinical information such as age and medication usage, all widely accessible in routine ICU care.

Although this feature set is larger than that of SOFA, we believe the substantial improvement in predictive performance and the ability to dynamically stratify risk justify the broader input space, particularly when automated extraction and calculation via EHRs are feasible. We have clarified this point in the revised manuscript.

Additional wording was added to the **discussion** section of our manuscript.

Discussion [page 19, lines 516-522]:

„In selecting variables for model development, we deliberately focused on features that are routinely available in most ICU settings and easily retrievable from standard electronic health records. This pragmatic approach was intended to ensure that the algorithm can be implemented across a wide range of clinical environments, without requiring access to highly specialized diagnostics or infrastructure. While the model includes more variables than traditional scores such as SOFA, this trade-off enables substantially higher predictive performance and dynamic risk assessment throughout the ICU stay.“

5. Will the stacked ensemble model further improve the model performance? Since the authors developed several ML models, did you try to use ensemble modeling to further improve the model performance (<https://doi.org/10.1016/j.jers.2021.12.003>)? There has been many empirical evidence that such ensemble models can help to achieve better accuracy than any individual model alone. At least you need to mention this potential.

Thank you for this thoughtful suggestion. We agree that ensemble models often enhance predictive performance. Indeed, the three best-performing algorithms in our comparison - LightGBM, XGBoost, and Random Forest - are all ensemble-based tree models that utilize bagging or boosting strategies.

Since these methods already combine multiple decision trees internally, the incremental benefit of adding an additional stacked ensemble is often limited, particularly when the individual models share similar structures and performance levels. Furthermore, stacked ensembles increase the complexity of

model development and maintenance, and may reduce interpretability - an essential factor for clinical application.

We acknowledge that combining fundamentally different model types (e.g., tree-based models with deep learning or probabilistic approaches) could provide greater complementarity and potential gains. Although we did not implement stacked ensembles in this study, we have included a **discussion** of this potential direction and its associated limitations in the revised manuscript.

Discussion [page 18, lines 500-506]:

„Although ensemble methods can theoretically improve performance, our best-performing models, including LightGBM, XGBoost, and Random Forest, are already ensemble-based approaches built on decision trees. Given their structural similarity, additional stacked ensembling is unlikely to yield substantial gains, while significantly increasing model complexity and limiting clinical interpretability⁴⁰. Future work could explore combining structurally different models, such as deep learning or probabilistic approaches, to assess whether heterogeneous ensembles offer practical advantages.“

6. It can be discussed further that while the model accuracy is important, but it may not translate to the clinical benefits such as reducing mortality in ICU. Thus, more work can be done to test the effectiveness of the model.

We fully agree with this important observation. While our model demonstrates excellent discrimination and calibration, predictive accuracy alone does not guarantee improved patient outcomes. The clinical utility of a model ultimately depends on how it is integrated into workflows, how clinicians respond to risk predictions, and whether those responses lead to meaningful changes in care.

We therefore emphasize that the LGBM-48h algorithm should be considered a decision-support tool, not a decision-maker. To assess its actual impact on ICU care and patient outcomes, such as reduced mortality, better triage, or optimized resource use, prospective clinical implementation studies are essential.

We have added a corresponding statement in the revised discussion and limitation sections, underlining the need for future work to evaluate the real-world effectiveness of the model.

Discussion [page 21, lines 590-597]

„While the LGBM-48h model demonstrated excellent discrimination and calibration across multiple settings, it is important to emphasize that predictive accuracy does not necessarily translate into clinical benefit. Risk prediction is only the first step in a broader chain of clinical decision-making. For a true improvement in patient outcomes such as reduced ICU mortality or improved resource allocation further steps are required, including seamless integration into clinical workflows, real-time availability, and appropriate clinical response to predictions. Future prospective implementation studies are needed to determine whether the use of dynamic mortality risk predictions can directly influence treatment strategies and improve outcomes in ICU settings.“

Limitation: [page 22, lines 609-613]

„Finally, although our model shows strong predictive performance, the impact on clinical outcomes such as reduced mortality or improved care processes has not yet been demonstrated. The present study does not evaluate how risk predictions influence clinical decisions or patient trajectories. Therefore, future prospective implementation studies are required to assess the real-world utility and effectiveness of the LGBM-48h algorithm in improving ICU outcomes.“

Reviewer 2:

Thank you for your assessment of our manuscript. Below, you will find our detailed response to your observation and comments, addressed point by point.

1. Why did the authors design the experiment to update every 24 hours to predict 48-hour mortality, rather than making predictions every hour, or every 6 or 12 hours, to predict 12-hour or 24-hour mortality? The authors should provide a clear rationale for this design choice.

We thank the reviewer for this thoughtful question. Our decision to generate predictions every 24 hours with a 48-hour prediction horizon was a deliberate balance between clinical utility, data availability, and operational feasibility.

In routine ICU practice, many clinical variables—especially laboratory values—are typically recorded only once per day. Using more frequent prediction intervals, such as hourly updates, would increase data sparsity and model noise, potentially necessitating artificial interpolation or imputation, which could compromise reliability.

Moreover, a key design goal was to develop a model that provides actionable, forward-looking risk information to support ICU decision-making without overburdening clinical staff. A 48-hour horizon offers sufficient lead time for planning interventions, while 24-hour update intervals align well with existing clinical workflows and rounds. This approach also reduces the frequency of system-generated alerts, helping to minimize alarm fatigue.

We have added a detailed explanation of this rationale in the **methods** section to clarify our choice of the 24-hour prediction interval and 48-hour horizon.

Methods [page 7, lines 176-183]:

„The model was designed to generate predictions once every 24 hours, with a 48-hour mortality prediction horizon. This decision reflects a pragmatic compromise between clinical applicability and data availability. In routine ICU practice, many input variables, particularly laboratory parameters, are recorded at most once per day. More frequent prediction intervals (e.g., every 6 or 12 hours) would result in increased data sparsity and may require imputation, potentially reducing model robustness. Furthermore, a 48-hour prediction window allows sufficient time for clinical teams to initiate interventions, while the 24-hour update cycle aligns with daily clinical workflows and reduces alert burden on healthcare staff.“

2. Please explicitly describe the inclusion and exclusion criteria for patient selection in the Method section.

Thank you for this valuable suggestion. To improve clarity, we have now summarized the full inclusion and exclusion criteria in a dedicated overview (see Supplemental Table 10).

A corresponding reference has been added in the revised **methods** section.

Methods [page 5, lines 122-124]:

„An overview of all inclusion and exclusion criteria applied for patient selection is provided in Supplemental Table 10.“

3. In line 206, the authors mention that the training dataset was used for tenfold cross-validation. Does this mean that tenfold cross-validation was only used for hyperparameter tuning? Why was the data split into equal-sized training and test datasets? Why not use tenfold cross-validation to partition training and test sets?

Thank you for raising this important point. We intentionally designed our data split and test strategy to reflect a realistic clinical development scenario. In practice, model development often begins with a fixed dataset, from which a dedicated training and test split is created, instead of a nested cross-validation as proposed by the reviewer.

To ensure adequate representation across all diagnostic subgroups, including smaller ones, we opted for a stratified 50:50 split. This approach guaranteed that even less prevalent diagnosis groups were sufficiently represented in both the training and test sets, preventing model bias and unstable subgroup estimates.

During model development, we performed tenfold cross-validation exclusively within the training set to tune hyperparameters and select the best-performing model. We acknowledge that a fully nested cross-validation scheme would offer a more rigorous estimate of generalization performance. However, implementing nested resampling in a stratified, diagnosis-balanced setting is computationally intensive and, more importantly, uncommon in applied clinical ML workflows.

As highlighted in the systematic review by Christodoulou et al. (BMJ, 2019, PMID: 30763612), most clinical prediction studies use simple train-test splits or non-nested cross-validation. We followed this convention to ensure interpretability, reproducibility, and clinical applicability, especially given the size and structure of our dataset. This point has been clarified in the revised Methods section.

We have clarified this rationale in the revised **methods** section.

Methods [page 9, lines 223-227]:

„This approach ensured sufficient representation of smaller diagnostic subgroups and avoided data sparsity in clinically relevant categories. While fully nested cross-validation would provide a more rigorous estimate, we opted for this simpler and more transparent strategy, consistent with clinical ML practice²⁷.“

4. I strongly agree with the authors' effort to stratify patient risk, as prediction is not the ultimate goal. The analysis regarding threshold selection is interesting. Although Supplemental Table 9 shows the distribution of risk group categories for patient stays, I find the presentation somewhat unintuitive. Could the authors consider visualizing how the model-triggered alarms were distributed during ICU stays for test patients, similar to Figure 6 in the article "Continuous prediction and clinical alarm management of late-onset sepsis in preterm infants using vital signs from a patient monitor"? This would provide a more intuitive understanding of alarm patterns.

We thank the reviewer for this valuable and constructive suggestion. We agree that visualizing the distribution of model-generated risk classifications over time can provide a more intuitive understanding of the algorithm's behavior throughout ICU stays.

We have created a new visualization that depicts the temporal distribution of predicted risk categories (high, intermediate, low) for a representative sample of ICU stays from the test dataset. This figure highlights the dynamics of the model's risk predictions over time and illustrates how alarm levels change during a patient's stay.

The new figure has been included in the Supplementary Information as **Supplemental Figure 2**, along with a corresponding explanation in the revised manuscript. We believe this addition enhances the communication of the model's temporal behavior and interpretability.

A reference to this figure has also been included in the revised **results** section.

Results [page 13, lines 364-370]:

„To provide a more intuitive understanding of alarm patterns and risk dynamics, we visualized the average predicted mortality probability and corresponding alarm levels over time for both ICU survivors and non-survivors (Supplemental Figure 2). Among non-survivors, a pronounced

nonlinear increase in risk was observed prior to death, accompanied by frequent transitions between risk categories. In contrast, survivors showed relatively stable predicted probabilities with predominantly low-risk alarms throughout their ICU stay.

5. In lines 323–327, the authors mention that the PPV remains low, indicating many false alarms. I would like to know the authors' thoughts on how to interpret or handle these false alarms.

We thank the reviewer for this important question. The relatively low PPV reflects the underlying class imbalance and the very low base rate of 48-hour mortality events in ICU data. In clinical prediction tasks with rare outcomes, this is a known trade-off when aiming for high sensitivity, especially in safety-critical settings where missing a critical event carries greater risk than issuing a false alert.

That said, we agree that false alarms must be managed carefully to prevent alarm fatigue and loss of clinical trust. Our approach enables threshold customization, which allows sites to balance sensitivity and PPV based on institutional needs and tolerable false alarm rates. Moreover, selecting an appropriate cut-off inherently requires balancing sensitivity and specificity, a well-known trade-off in clinical prediction modeling. In our study, we deliberately chose to prioritize high sensitivity, accepting a lower PPV in order to minimize the risk of missing critical deterioration events. Importantly, false positives do not necessarily equate to incorrect alerts: it is conceivable that some flagged patients were indeed in clinically unstable conditions but did not meet the endpoint of death within 48 hours. The optimal threshold depends on the intended clinical use, the local hospital workflow, and physician judgment. Therefore, the choice of cut-off is context-dependent and should be aligned with the specific goals of deployment - whether for early warning, closer patient monitoring, or structured case reviews.

Notably, our study did not yet evaluate how false alarms translate into actual clinical consequences or actions. Understanding how alerts influence behavior, decision-making, and ultimately patient outcomes would be a critical next step to assess the practical utility of the algorithm beyond its technical performance.

Therefore, we have added a corresponding paragraph in the **discussion** to reflect our interpretation and outline strategies for future alarm management, including the potential for combining model predictions with clinician input or suppression strategies (e.g., alarm silencing windows after high-risk calls).

Discussion [page 20, lines 544-556]:

„In addition, the relatively low positive predictive value (PPV) observed in our model reflects the inherent trade-off between sensitivity and specificity when predicting rare events such as 48-hour ICU mortality. In this study, we intentionally favoured high sensitivity to minimize the risk of missing critical deterioration, accepting a lower PPV as a consequence. While this increases the number of false positives, many of these flagged patients may still have been clinically unstable and thus benefited from increased monitoring. Alarm fatigue is a well-documented phenomenon in intensive care settings, where frequent non-actionable alerts can desensitize staff, delay responses, and jeopardize patient safety⁴⁵. To mitigate this, our model supports customizable thresholds, allowing hospitals to calibrate sensitivity and PPV based on clinical context, institutional workflow, and risk tolerance. Moreover, the optimal threshold should reflect the specific clinical goal whether early warning, triage support, or case prioritization and ideally involves clinical judgment alongside algorithmic predictions. Furthermore, future studies should investigate the downstream effects of model-generated alerts on clinical decisions and outcomes.“

6. In Figure 4b, what about the results for the intermediate and low-risk groups? Also, in Figure 4c, should patient 1 be labeled as deceased and patient 2 as survived?

We thank the reviewer for this important suggestion. To address this, we have expanded the graphical analysis of Figure 4b and now provide a separate supplemental figure illustrating the temporal dynamics of predicted risk across all three risk categories (low, intermediate, and high). We have added a

corresponding explanatory sentence to the **results** section to reference this new supplementary information material.

Also, we thank the reviewer for pointing out the labeling issue in Figure 4c. Indeed, the labels for patient 1 and patient 2 were mistakenly inverted. We have corrected this in the revised version of the figure, where patient 1 is now labeled as deceased and patient 2 as survived.

Results [pages 14, lines 391-394]:

*„This association was also evident for the low- and intermediate-risk groups, with the GAM-derived mortality curves demonstrating a continuous increase in mortality for longer time spent in the intermediate-risk category, and a corresponding decrease for increasing exposure to the low-risk category (**Supplemental Figure 3**).“*

7. I suggest the authors include the AUC values directly in ROC plots such as Figures 3a, 5c, and 5d. This would make it easier for readers to view the AUC results without referring back to the main text.

Thank you for this helpful suggestion. We agree that including the AUC values directly in the ROC plots enhances readability and interpretability. Accordingly, we have updated Figures 3a, 5c, and 5d to display the respective AUROC values within the plots. These updated versions have been included in the revised PDF/JPEG of the figures.

8. The authors included a summary at the end of almost every subsection in the Results. Please note that the Results section should objectively present experimental findings without summarizing or interpreting them. Any interpretations or summaries should be moved to the Discussion section.

We thank the reviewer for this valuable editorial comment and fully agree with the recommendation. We carefully revised the Results section and removed all sentences that contained interpretative elements or speculative wording. Where needed, formulations were adapted to ensure a strictly descriptive and objective reporting of the experimental findings. All content with interpretative character was either rephrased to be purely descriptive or removed entirely.

Additional Editorial Revisions

In addition to the revisions made in response to the reviewers' comments, we implemented the following editorial changes in the revised manuscript:

1. Data and Code Availability

We added a statement on data and code availability at the methods section, in accordance with Communications Medicine's policies. Specifically, we clarify that both the data and code supporting the findings of this study are available from the corresponding author upon reasonable request and subject to institutional and ethical restrictions.

Methods [page 5, lines 124-125]:

"The data and the codes that support the findings of this study are available from the corresponding author upon reasonable request and subject to institutional and ethical restrictions."

2. Reference Formatting

All references have been reformatted to comply with the Nature referencing style, as required by the journal. We ensured consistency throughout the reference list, including citation style for journal articles, online sources, and software tools.

3. Figure File Adjustments

In accordance with the journal's submission guidelines, we removed the word "Figure" from all main figure file names (Figures 1–5), so that only the numbering remains (e.g., "a", "b", etc.). Captions have been kept in the manuscript file as required.